# Clinical Remission Using Personalized Low-Dose Intravenous Infusions of *N*-acetylcysteine with Minimal Toxicities for Interstitial Cystitis/Bladder Pain Syndrome

**DOI:** 10.3390/jpm11050342

**Published:** 2021-04-24

**Authors:** Dipnarine Maharaj, Gayathri Srinivasan, Sarah Makepeace, Christopher J. Hickey, Jacqueline Gouvea

**Affiliations:** 1Department of Medicine, The Maharaj Institute of Immune Regenerative Medicine, Boynton Beach, FL 33437, USA; education@miirm.org (G.S.); sjmakepeace9@gmail.com (S.M.); jgouvea@bmscti.org (J.G.); 2Palm Beach Atlantic University, West Palm Beach, FL 33401, USA; chris_hickey@pba.edu

**Keywords:** interstitial cystitis, bladder pain syndrome, *N*-acetylcysteine, personalized medicine, pro-inflammatory cytokines, inflammation, TNF-α, IL-1β, anti-inflammatory therapy

## Abstract

Interstitial Cystitis or Bladder Pain Syndrome (IC/BPS) is a heterogeneous condition characterized by elevated levels of inflammatory cytokines, IL-1β, IL-6, IL-8, IL-10, TNF-α, and is associated with debilitating symptoms of pelvic pain and frequent urination. A standard of care for IC/BPS has not been established, and most patients must undergo a series of different treatment options, with potential for severe adverse events. Here, we report a patient with a 26-year history of IC/BPS following treatment with multiple therapies, including low doses of etodolac, amitriptyline and gabapentin, which she was unable to tolerate because of adverse effects, including headaches, blurred vision and cognitive impairment. The patient achieved a complete clinical remission with minimal adverse events after 16 cycles of *N*-acetylcysteine (NAC) intravenous (IV) infusions over a period of 5 months, and pro-inflammatory cytokine levels were reduced when compared to measurements taken at presentation. Personalized low dose NAC IV infusion therapy represents an effective, safe, anti-inflammatory therapy administered in the outpatient setting for IC/BPS, and warrants further investigation.

## 1. Introduction

Interstitial Cystitis (IC), or Bladder Pain Syndrome (BPS), is a heterogeneous condition involving inflammation of the urinary bladder wall that produces debilitating symptoms of pelvic pain and frequent urination in affected patients. Since the etiology of IC/BPS is yet unknown, it is clinically diagnosed by exclusion in patients exhibiting bladder or pelvic pain, urinary frequency, urgency, and nocturia [1,2,3,4]. Another consequence of the imprecise diagnostic criteria is a lack of certainty in IC/BPS occurrence. It has been established that the condition is disproportionately seen among women [5,6], comprising up to 90% of cases in the United States [6]. A 2016 report estimated that between 2.7% and 6.5% of American women experience IC/BPS symptoms, but prevalence in both men and women is likely underreported [6]. The heterogeneous nature of symptoms may be explained by the increasingly widespread theory that IC/BPS is likely caused by multiple diseases which produce chronic inflammation, manifesting as various lower urinary tract symptoms [1].

A standard of care for interstitial cystitis has not yet been established and, as a consequence, most patients must undergo a series of different treatment options to elucidate what is effective for their individual situation. Current treatments for IC/BPS include drug therapy such as dimethyl sulfoxide, amitriptyline, pentosane polysulfate, cyclosporine A, or nonsteroidal anti-inflammatory compounds, as well as alternative medicine treatments involving the modification of lifestyle factors [2,7,8]. Although pharmacological therapies have shown some benefit in control of IC/BPS symptoms, they may have a negative impact on the patient due to the inherent potential of adverse events [9]. In vivo studies using mouse models have shown that pentosan polysulfate may cause uncontrolled activation of immune function, which may lead to chronic inflammation, hypersensitivity, or autoimmunity [10]. As IC/BPS therapies continue to evolve, triple-targeted therapy using gabapentin, amitriptyline, and a nonsteroidal anti-inflammatory drug (NSAID) have demonstrated improved clinical outcomes [11,12]. Gabapentin is an effective anticonvulsant which addresses the chronic pain, including neuropathic pain, associated with IC/BPS [12]. Serving as an extension of pain management, amitriptyline, a tricyclic antidepressant, has been found to act as an effective analgesic in many patients [12,13]. Using these two drugs in combination with NSAIDs is a strategy for treatment of IC/BPS that attempts to target pain and overactive bladder symptoms, including voiding frequency and urgency [11], in a more effective way than using one medication alone.

Despite promising results, triple-targeted therapies are not without potential for severe adverse events. Gabapentin carries the risks of psychedelic properties, dizziness, and respiratory failure [14], while amitriptyline can cause serious side effects such as blurred vision, tachycardia, and hypotension [15]. In combination, studies show that these drugs may cause side effects such as dizziness and sedation in up to half of participants [16]. Very few studies have examined the effects of this triple-therapy for IC/BPS treatment, and those that have been conducted focused on small cohorts with limited follow-up periods [11,12]. Additionally, these studies show that, although alleviation of symptoms was reported early in treatment with triple-therapy, the initial decrease in symptoms levels off after weeks or months of treatment [11,12].

*N*-acetylcysteine (NAC) is a potential alternative novel therapy to reduce the inflammation associated with IC/BPS. NAC works directly by scavenging free radicals, and indirectly by increasing intracellular levels of glutathione, the most prevalent cellular antioxidant [17]. NAC has been shown to reduce oxidative stress in cells [18], as well as exert systemic anti-inflammatory effects in the body, with little to no harmful effects reported [17]. Clinically, NAC is used to treat a wide variety of diseases and conditions, including several inflammation-related diseases such as asthma, ulcerative colitis, and chronic bronchitis [17]. Taken together, the dualistic antioxidant and anti-inflammatory functions of NAC can be utilized to treat patients with IC/BPS by modifying the immune microenvironment to augment anti-inflammatory responses.

Although outcome measurements of IC/BPS treatments are largely symptom-based, multiple studies have reported the value of tracking cytokines in monitoring the disease course [19,20]. Expressions of IL-1β, IL-6, IL-8, and TNF-α are all significantly elevated in animals [21], as well as human patients with IC/BPS, in comparison to individuals without the disease [19,20]. These persistently increased pro-inflammatory cytokines indicate a chronic inflammatory immune response, which is a likely contributor to IC/BPS pathology and symptoms [19,20,21].

We use a novel personalized treatment approach of NAC intravenous (IV) infusion dosing and frequency based upon serial plasma cytokine measurements to monitor the efficacy of treatment by reduction of symptoms and cytokine levels, while minimizing adverse events. In this case report, we describe a female patient presenting with IC/BPS for over 26 years following multiple therapies, including low-dose triple therapy of amitriptyline, gabapentin, and NSAID, which were terminated due to adverse events. At our clinic, the patient achieved a complete clinical remission with minimal adverse events after 16 cycles of NAC IV infusions over a period of 5 months, and pro-inflammatory cytokine levels were reduced when compared to measurements taken at presentation.

## 2. Results

### Case Report

A 70-year-old woman with a history of IC/BPS first attended our clinic in September 2019 (timeline of the key information is summarized in Figure 1). In 1961, the patient had frequent urination with recurrent urinary tract infections (UTI). In 1968, the patient was involved in a motor vehicle accident and she had fractures of L3-L5 and injury to her bladder. She subsequently developed chronic bladder and pelvic pain, urinary frequency, urgency and nocturia, and she was diagnosed with IC/BPS in 1994. The patient opted not to take prescribed medications and she adopted a holistic lifestyle consisting of a vegan diet, acupuncture, and massage therapy. In 2012 the patient was enrolled in an IC/BPS medical trial which included low doses of etodolac, amitriptyline and gabapentin for 3 months. After experiencing adverse effects including headaches, blurred vision and cognitive impairment, the patient ended her participation in the clinical trial and she resumed her holistic program.

In September 2019, the patient attended our clinic for evaluation of her disease and immune status. The patient reported mild urinary incontinence, difficulty starting and frequent, painful urination. A Pelvic Pain and Urgency/Frequency (PUF) questionnaire was administered to rule out other probable causes of painful urination such as endometriosis, or UTI. The PUF questionnaire was chosen over other IC/BPS questionnaires due to its comprehensive nature and its demonstrated ability to distinguish IC/BPS more efficiently [22,23]. The PUF scale showed a score of 28. Scores greater than 10 are associated with 90% sensitivity for IC/BPS [24]. Plasma cytokines showed levels of IL-1β at 19.49 pg/mL (elevated at 50th–75th percentile), TNF-α at 10.54 pg/mL (elevated at 75th–90th percentile), IL-10 at 7.72 (elevated at 50th–75th percentile), IL-8 at 2.24 pg/mL (at 25th–50th percentile), and IL-6 at 3.02 pg/mL (elevated at 50th–75th percentile) (Figure 2). Reference ranges by percentile were determined by calculating the mean and standard deviation from a cohort of healthy men and women. Treatment options discussed included responses and side effects of pentosane polysulfate, amitriptyline, cimetidine, cyclosporin A, hydroxyzine, self-care practices and behavior modification techniques, or no therapy. The patient refused treatments with potentially severe side effects and continued her holistic program.

In February 2020, the patient reported persistent urinary incontinence and worsening frequent/painful urination. The patient gave informed consent for treatment using personalized low dose *N*-acetylcysteine (NAC) IV infusions, including publication of results. The objective was to treat the chronic inflammation associated with IC/BPS using weekly low dose NAC IV infusions. Variation of the duration, dosing and frequency of administration of NAC was based on the patient’s symptoms and plasma cytokine levels.

On 5 May 2020, the patient started her weekly NAC IV infusions at a low dose of 6000 mg per IV infusion cycle. Plasma cytokines levels were elevated, with IL-1β at 18.27 pg/mL (elevated at 50th–75th percentile), TNF-α at 6.22 pg/mL (elevated at 50th–75th percentile), IL-10 24.72 pg/mL (elevated above 90th percentile), IL-8 at 1.79 pg/mL (at 25th–50th percentile), and IL-6 at 3.37 pg/mL (elevated at 50th–75th percentile) (Figure 2). On 26 May 2020, the patient complained of bladder discomfort and urinary frequency, and urinalysis confirmed an *Escherichia coli* infection for which the patient was treated with 500 mg of levofloxacin once a day for three days, along with her weekly NAC IV infusions. By 1 June 2020, the patient denied any urinary frequency, urgency, or bladder discomfort.

On 10 June 2020, the patient had successfully completed 5 NAC IV infusions with no complications. The patient was asymptomatic of all pain and urinary symptoms, and she chose to suspend NAC treatment. On 17 July 2020, the patient reported urinary urgency and she resumed weekly NAC IV infusions. After three more IV infusions the patient reported that her urinary urgency, incontinence, pains and spasms had completely resolved, and the patient remarked that this was the first time that she was completely pain free.

The patient experienced no associated side effects of the treatment. Plasma cytokine evaluation showed reductions in IL1-β at 10.75 pg/mL (at 25th–50th percentile), TNF-α at 3.56 pg/mL (at 25th–50th percentile), IL-10 8.00 pg/mL (elevated at 50th–75th percentile), IL-8 at 1.02 pg/mL (below 10th percentile), and IL-6 at 2.58 pg/mL (at 25th–50th percentile) (Figure 2). The patient was recommended to continue weekly NAC IV infusions to further reduce chronic inflammation.

In October 2020, the patient completed her 16th and final NAC IV infusion cycle, with no associated side effects. The patient continued to report no recurrence of urinary pains, spasms, urgency, or incontinence. Plasma cytokines showed continued improvements, with IL1-β at 7.88 pg/mL (at 25th–50th percentile), TNF-α at 1.45 pg/mL (below 10th percentile), IL-10 at 3.25 pg/mL (at 10th–25th percentile), IL-8 at 1.09 pg/mL (below 10th percentile), and IL-6 at 4.22 pg/mL (elevated at 50th–75th percentile) (Figure 2). The PUF questionnaire given at this time yielded a score of 11, which was an improvement from the baseline of 28. Five months following completion of the NAC IV infusion cycles, the patient remains in complete clinical remission. This is the longest duration of time that the patient has remained completely symptom-free.

## 3. Discussion

We report a case of complete clinical remission of IC/BPS with personalized NAC IV infusions in a patient who had suffered for at least 26 years with debilitating symptoms, despite undergoing a series of commonly used treatments. The patient did not experience any therapy-related adverse events throughout the course of personalized NAC treatment. Symptom relief occurred after 4 NAC IV infusions, and the patient achieved complete clinical remission after completing 16 NAC IV infusions, with marked improvements in levels of plasma cytokines.

NAC as a treatment for interstitial cystitis is a novel approach. Based on experimental data and successful treatment of several chronic inflammation-related diseases [17], we personalized the dose and schedule of low-dose NAC by monitoring symptoms and plasma cytokines before and after infusions. Recent studies using NAC to treat liposaccharide-induced interstitial cystitis in rats demonstrated positive results [25,26]. Intraperitoneal NAC injections in rat subjects reduced inflammation, reversed the progression of fibrosis, and restored the integrity of the urothelium, resulting in improvement of abnormal voiding symptoms [25]. Although it is generally well-tolerated with minimal adverse effects [18], NAC may act like a pro-oxidant at high doses, increasing oxidative stress and stimulating inflammation and abnormal cell proliferation [25,27,28]. Because of this potential risk, appropriate dosing is crucial in clinical application of NAC. In our treatment plan, variation of duration, dosage, and frequency of administration of NAC was based on plasma cytokine levels, and the patient was monitored frequently, with symptom tracking using the Pelvic Pain and Urgency/Frequency (PUF) questionnaire. The patient received low doses of 6000 mg per IV infusion cycle and showed no adverse reactions to the treatment. Although there is not a universal standard for NAC dosage, one meta-analysis showed ranges from 500–1200 mg once or twice per day, to 14,000 mg in multiple IV infusions over the course of a day [29]. A recent study using NAC to treat the cytokine storm of COVID-19 used a dosing regimen of 30,000 mg in three separate doses over 24 h, with favorable results [30].

Chronic inflammation manifesting as various lower urinary tract symptoms is characteristic of IC [1,3,4]. Elevated levels of cytokines such as IL1-β, TNF-α, IL-8, and IL-6 in patients with IC/BPS can serve as a guide for the dosing of NAC. IL-1β is a pro-inflammatory cytokine that is primarily increased in patients with IC/BPS, and animal model studies have shown that NAC inhibits the pro-inflammatory expression of IL-1β [19,20,21]. Immediately prior to beginning treatment in May 2020, the patient exhibited elevated levels of IL-1β at 18.27 pg/mL (Figure 2a), when the patient’s symptoms of urinary pain, incontinence and urgency were severe, and accompanied by a high PUF score of 28. With continuous treatments of NAC, IL-1β was reduced to 7.88 pg/mL, which was consistent with the resolution of symptoms and a decrease of the PUF score to 10.

TNF-α is a pleiotropic, pro-inflammatory cytokine released by mast cells in the bladder urothelium of patients with IC, and this overexpression has been shown to participate in IC/BPS pathology [31]. In a study using transgenic mice, TNF-α was shown to promote urothelial apoptosis and cause the characteristic symptoms of pelvic pain, voiding abnormalities and urothelial lesions [31]. Our patient showed elevated TNF-α levels prior to treatment, at 6.22 pg/mL (Figure 2b), higher than the average value reported in IC/BPS patients of 2.63 pg/mL [19,20]. After completing the full 16 cycles of NAC IV infusions, the patient’s levels decreased to 1.45 pg/mL and correlated with symptom resolution. This value is much closer to the average TNF-α expression observed in control patients of 0.91 pg/mL [19,20].

Following completion of the 16 IV infusions of NAC in November, 2020, the patient’s IL-8 levels decreased, indicating improvement (1.09 pg/mL vs. 1.79 pg/mL) (Figure 2d), and IL-10 levels also decreased (3.25 pg/mL vs. 24.72 pg/mL) (Figure 2c). While IL-10’s anti-inflammatory properties are well documented, studies from recent years have shown its potential pro-inflammatory characteristics [32]. Patients treated with IL-10 displayed increased levels of inflammatory markers CRP, IL-6, and lipopolysaccharide binding protein (LBP) [33]. Other studies showed that IL-10 administration was associated with increases in levels of IL-6, IL-8 and TNF-α, although these did not show statistical significance [34]. This is consistent with the findings in our patient who demonstrated increased levels of IL-10, which correlated with increased levels of IL-8 and TNF-α; these levels decreased when IL-10 expression decreased. Additional studies have revealed that IL-10 is significantly elevated in patients with UTIs [35,36]. Our findings are consistent, since, at the time of our patient’s UTI at the beginning of treatment (Figure 1), she had the highest levels of IL-10 (Figure 2c). Once her UTI symptoms were resolved, IL-10 expression showed a dramatic decrease (Figure 2c). Collectively, these studies and our findings support the pro-inflammatory characteristics and increased expression of IL-10 in inflammatory environments [32,33,34,35,36].

Interestingly, IL-6 levels initially decreased, from 3.37 pg/mL to lowest levels of 2.58 pg/mL in July after 5 NAC IV infusions, but then increased to 4.22 pg/mL in October 2020. The reason for the subsequent increase in IL-6 levels while on NAC is unclear, since all other pro-inflammatory cytokines were reduced and she displayed no symptoms. Our patient’s older age may be a factor, since younger patients with IC/BPS treated with cyclosporin A or pentosane polysulfate sodium did not exhibit significant changes to IL-6 levels; however, in older patients (patients older than 53 years), IL-6 levels were reduced after cyclosporin A treatment [37]. Further serum immune testing performed to monitor IL-6 in December 2020 showed a decreased level of 2.90 pg/mL, which was consistent with the decreases seen in her other pro-inflammatory cytokines. We are continuing to monitor our patient’s IL-6 levels after completion of treatment. Overall, our patient’s course of symptom relief correlated with an improvement of plasma cytokines levels. In July 2020, the patient reported a complete absence of urinary pains and spasms, as well as urinary urgency and incontinence. The patient has remained in clinical remission for five months following the completion of the IV infusion cycles in October 2020, and she has remained off all treatment.

This report describes a previously unreported approach for treating IC/BPS by measuring inflammatory biomarkers IL-1β, TNF-α, IL-8, IL-10 and IL-6, and tailoring the dose and frequency of NAC IV infusions in response to the reduced expression of these biomarkers, combined with symptom monitoring using the PUF score. Our report serves as a basis for a precise and personalized low toxicity anti-inflammatory therapeutic option in patients with debilitating IC/BPS symptoms who have been unresponsive to commonly used treatments.

## 4. Conclusions

IC/BPS is associated with debilitating symptoms which often show short-lived responses or lack of response to commonly used treatments. Our data showed that personalized treatment with low-dose IV infusions of NAC was associated with reduced expression of pro-inflammatory cytokines of IC/BPS, and resulted in the desired outcome of complete remission with no adverse events.

## 5. Future Perspective

This case report shows that personalized anti-inflammatory treatment with NAC IV infusions resulted in complete clinical remission, with no adverse events, in a patient with IC/BPS who had suffered with chronic debilitating symptoms for over 26 years and who had a poor response and adverse events to commonly used treatments. This unique approach is an important contribution and it supports the need for future research concerning this therapeutic option to improve outcomes and minimize side effects in patients with IC/BPS.

## Figures and Tables

**Figure 1 jpm-11-00342-f001:**
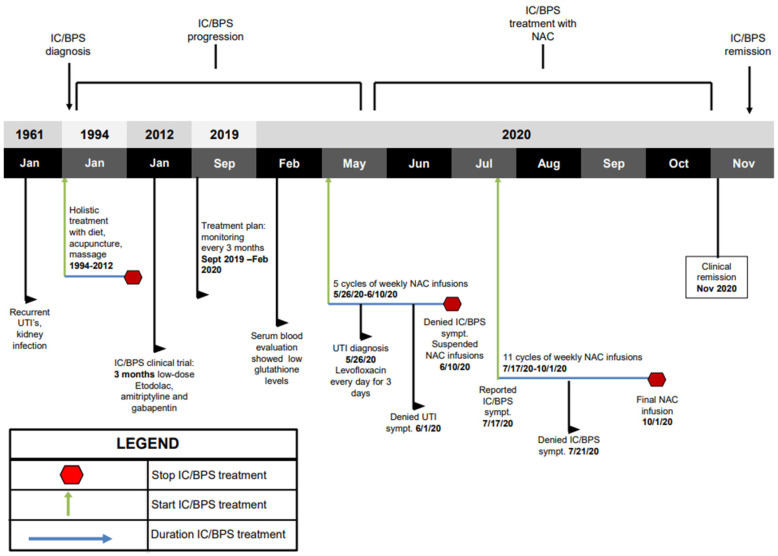
Key timeline information. Green arrows indicate treatment initiation. Blue arrows indicate the duration of treatment. Red octagons indicate when the patient stopped treatment. Beginning in 1994 after IC/BPS diagnosis, holistic treatment was used. In 2012, patient underwent a 3-month IC/BPS clinical trial using low-dose triple therapy. In May 2020, patient began weekly 6000 mg NAC IV infusions. Patient suspended IV infusions in June 2020. Patient resumed NAC IV infusion treatment in July 2020. NAC treatment was completed in October 2020.

**Figure 2 jpm-11-00342-f002:**
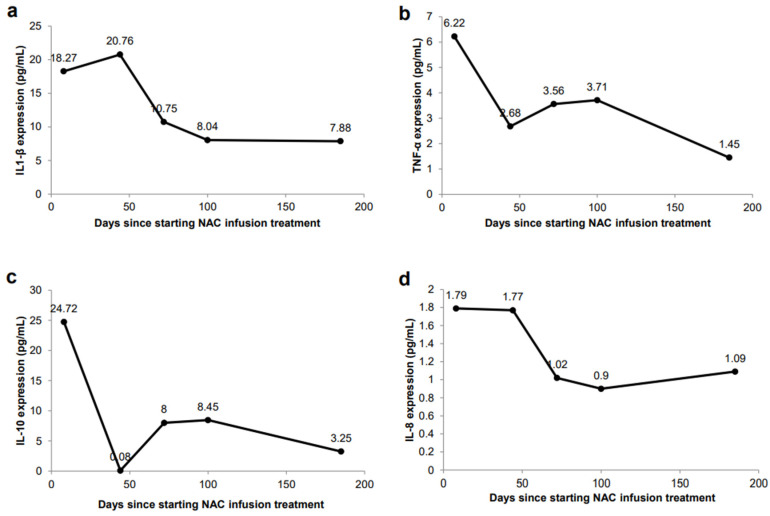
Changes in plasma cytokine secretion in response to NAC IV infusions. (**a**) IL-1β expression (ELISA) in peripheral blood. (**b**) TNF-α expression (ELISA) in peripheral blood. (**c**) IL-10 expression (ELISA) in peripheral blood. (**d**) IL-8 expression (ELISA) in peripheral blood.

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
