# Peer review of "Clinical Remission Using Personalized Low-Dose Intravenous Infusions of N-acetylcysteine with Minimal Toxicities for Interstitial Cystitis/Bladder Pain Syndrome"

_jpm, 2021, doi:10.3390/jpm11050342_

Round 1
Reviewer 1 Report
Dear Authors,
The manuscript tries to identify a treatment plan with NAC for IC/BPS, from a lack of clear understanding of the disorder's etiology, symptoms variation across patients, and a paucity of high-quality data regarding IC/BPS treatments' efficacy and safety. The manuscript is well written and organized.
Here are my comments:
- Can you please explain what you understand by "NAC infusion"? It is not clear. Do you mean intravesical instillation?
- Please explain why you use the Pelvic Pain and Urgency/Frequency (PUF) questionnaire and not the validated IC Symptom and Problem Index or the Genitourinary Pain Index for evaluating the patient.
- Please include the following references in your manuscript (AUA guidelines for IC/BPS):
Hanno PM, Burks DA, Clemens JQ, et al. AUA guideline for the diagnosis and treatment of interstitial cystitis/bladder pain syndrome. J Urol. 2011;185(6):2162-2170. doi:10.1016/j.juro.2011.03.064
Hanno PM, Erickson D, Moldwin R, Faraday MM; American Urological Association. Diagnosis and treatment of interstitial cystitis/bladder pain syndrome: AUA guideline amendment. J Urol. 2015;193(5):1545-1553. doi:10.1016/j.juro.2015.01.086
Author Response
Dear Editor and Reviewer,
The authors of the case report titled, “Clinical Remission using Personalized Low-Dose Infusions of N-Acetyl Cysteine with Minimal Toxicities for Interstitial Cystitis/Bladder Pain Syndrome” would like to extend our gratitude to the reviewers for their attention to our manuscript. Please find below our responses to the Reviewer 1 comments.
Reviewer 1:
Comment 1: Can you please explain what you understand by "NAC infusion"? It is not clear. Do you mean intravesical instillation?
Response:
We agree with Reviewer 1 that NAC infusions should be clarified. We modified the Abstract in Line 17 to read: “The patient achieved a complete clinical remission with minimal adverse events after 16 cycles of N-acetylcysteine (NAC) intravenous (IV) infusions over a period of 5 months and the pro-inflammatory cytokine levels were reduced when compared to measurements taken at presentation.” The remainder of the manuscript where “infusions,” are mentioned has subsequently been modified to specify that IV infusions were administered.
Comment 2: Please explain why you use the Pelvic Pain and Urgency/Frequency (PUF) questionnaire and not the validated IC Symptom and Problem Index or the Genitourinary Pain Index for evaluating the patient.
Response 2:
We used the Pelvic Pain and Urgency/Frequency (PUF) questionnaire instead of the IC Symptom and Problem Index due to the questions being more comprehensive. The PUF has additional items related to pelvic pain and pain related to sexual intercourse. The PUF has also been shown to be more efficient than the IC Symptom and Problem Index. In the article “Efficiency of Questionnaires Used to Screen for Interstitial Cystitis,” Kushner and Moldwin examined the ability of the PUF score and the ICSI/ICPI to distinguish IC from other urinary tract pathologies in a group of 220 patients at a urology clinic. All 3 scales did distinguish IC, and the PUF score (≥ 13) did so more efficiently. However, none of the tools had sufficient specificity to serve as a sole diagnostic. The authors recommended the use of the ICSI, ICPI, UW-IC Scale, or PUF Score to assist with diagnosis and follow response to therapeutic intervention in patients with IC. The PUF was used in our patient because of the additional items related to pelvic pain and pain related to sexual intercourse and due to its increased efficiency.
In our attempt to clarify the point, we modified Results Line 120 to read: “The PUF questionnaire was chosen over other IC/BPS questionnaires due to its comprehensive nature and its demonstrated ability to distinguish IC/BPS more efficiently [22, 23].”
Comment 3: Please include the following references in your manuscript (AUA guidelines for IC/BPS):
Hanno PM, Burks DA, Clemens JQ, et al. AUA guideline for the diagnosis and treatment of interstitial cystitis/bladder pain syndrome. J Urol. 2011;185(6):2162-2170. doi:10.1016/j.juro.2011.03.064
Hanno PM, Erickson D, Moldwin R, Faraday MM; American Urological Association. Diagnosis and treatment of interstitial cystitis/bladder pain syndrome: AUA guideline amendment. J Urol. 2015;193(5):1545-1553. doi:10.1016/j.juro.2015.01.086
Response 3:
To include these suggested references, we modified the Introduction Line 30 and the Discussion Line 202. They are listed as References 3 and 4, respectively.

Reviewer 2 Report
Thank you for considering me as a reviewer for this publication. I have provided my comments as follows.
General Comments:
In my opinion, this article is very well written and contributes to the existing knowledge about interstitial cystitis/painful bladder syndrome.
I could not find any logical errors in the presentation and the approaches used.
Interstitial cystitis/painful bladder syndrome is very often, but it is overlooked. It has a major influence on the quality of life with serious consequences. Sometimes it can be very challenging to establish the right diagnosis. The efficacy of treatment is often poor and authors present different methods of treatment, with emphasis on novel therapy with N-acetylcysteine in this case.
In the discussion section, all components are critically reviewed.
This article will be useful for the urologist, nephrologist, neurologist, and other physicians that treat voiding disorders.
Specific Comments: none.
Author Response
Dear Editor and Reviewer,
The authors of the case report titled, “Clinical Remission using Personalized Low-Dose Infusions of N-Acetyl Cysteine with Minimal Toxicities for Interstitial Cystitis/Bladder Pain Syndrome” would like to extend our gratitude to the reviewer for their attention to our manuscript. This reviewer did not have any comments that required our response.
